# Interaction between Pharmaceutical Drugs and Polymer-Coated Fe_3_O_4_ Magnetic Nanoparticles with Langmuir Monolayers as Cellular Membrane Models

**DOI:** 10.3390/pharmaceutics15020311

**Published:** 2023-01-17

**Authors:** Sara Natalia Moya Betancourt, Candelaria Inés Cámara, Julieta Soledad Riva

**Affiliations:** 1Consejo Nacional de Investigaciones Científicas y Técnicas (CONICET), Instituto de Fisicoquímica y Química Inorgánica de Córdoba (INFIQC), Córdoba 5000, Argentina; 2Departamento de Fisicoquímica, Facultad de Ciencias Químicas, Universidad Nacional de Córdoba, Haya de la Torre y Medina Allende, Ciudad Universitaria, Córdoba 5000, Argentina

**Keywords:** magnetic nanoparticles, chitosan, diethyl amino-ethyl dextran, triflupromazine, diclofenac, biomembrane models, Langmuir monolayer, maximum insertion pressure

## Abstract

Surface modification of magnetic nanoparticles (MNPs) has been reported to play a significant role in determining their interactions with cell membranes. In this research, the interactions between polymer functionalized (chitosan, CHI or diethylamino-ethyl dextran, DEAE-D) Fe_3_O_4_ MNPs, pharmaceutical drugs and model cell membranes were investigated by Langmuir isotherms and adsorption measurements. In this study, 1,2-distearoyl-sn-glycerol-3-phosphate (DSPA) phospholipid monolayers were used as cell membrane models. Insertion experiments demonstrate that diclofenac (DCFN) is not absorbed at the air–water interface, whereas triflupromazine (TFPZ) has a MIP (maximum insertion pressure) of 35 m Nm^−1^. The insertion of composites MNPs:TFPZ or DCFN has larger MIP values, indicating that the MNPs are adsorbed on the monolayer with the drugs. An Fe_3_O_4_@CHI:DCFN composite presented an MIP of 39 m Nm^−1^ and Fe_3_O_4_@DEAE-D:DCFN presented an impressive MIP of 67 mNm^−1^. In the case of TFPZ, the enhancement in the MIP values is also evident, being 42 mNm^−1^ for Fe_3_O_4_@CHI:TFPZ and 40 mNm^−1^ for Fe_3_O_4_@DEAE-D:DCFN composite. All MNPs:drugs composites have MIP values greater than commonly accepted membrane pressure values, indicating that MNPs:drugs can penetrate a cellular membrane. The fact that the composite MNPs:drugs present greater MIP values than separated compounds indicates that polymer-coated MNPs can act as good drug delivery systems.

## 1. Introduction

Pharmacologically active compounds are usually amphiphilic or hydrophobic molecules whose site of action in the organism is frequently the plasma membrane. These compounds undergo different types of interaction with cell membranes, and even if their target is intracellular, the interaction with this first barrier plays a fundamental role [1].

Many membrane–drug interactions have been elucidated using the Langmuir monolayer technique, either to verify hypotheses about their mode of action, to determine their toxicity or to select their molecular targets [1]. Generally, these issues were impossible to explain from the results obtained in living cells due to their complexity. In contrast, using Langmuir monolayers, the affinity of a drug for a particular membrane component can be selectively ascertained by examining their mutual interactions. The simplest model is a one-component monolayer, which, although a simplification, allows easy characterization of the process used to target a drug to the cell and its potential toxicity. In this context, many drugs with different chemical structures and pharmacological effects have been investigated, such as antimicrobial [2,3], antitumoral [4,5], anti-inflammatory [6,7,8,9], anesthetic [10], antipsychotic [11,12,13], cardiac [14] and antihistaminic drugs [15]. Additionally, the study of drugs with monolayers of liquids is interesting, since recently, lipid miscellas encapsulating different drugs have been proposed as drug delivery systems. In this context, recent works have proposed Pluronic F127 nanomicelles to encapsulate and control the release of the antineoplastic drug carfilzomib [16], nanostructures to deliver therapeutic drugs for pancreatic cancer treatment via the EPR-mediated effect [17] and a pH-sensitive vesicular system of non-ionic surfactants and cholesterol loaded with deferasirox, an iron chelator used for the treatment of transfusional iron [18].

Among all the anti-inflammatory non-steroidal drugs, diclofenac DCFN (see Figure 1c) has been widely studied in monolayers due to it being able incorporate into membranes, affecting the lipidic organization and leading to solubilization of the structure [6]. Sousa et al. studied the interaction between diclofenac and soya phosphatidylcholine (SP) using different methodologies, co-spread with SP and penetrating a pre-formed monolayer [6]. DCFN was found to penetrate the hydrophobic part of the phospholipid monolayer and the isotherms become more expanded as the DCFN concentration increases.

Regarding anti-psychotic drugs, triflupromazine (TFPZ) acts at the central nervous system, blocking dopamine receptors. It is an amphiphilic molecule with a hydrophobic phenothiazine ring and hydrophilic ionizable amine tail group (see Figure 1d). Due to its chemical structure, it can be partitioned into biological membranes inducing several lateral effects. For this reason, several authors have studied the interaction of TFPZ with zwitterionic and anionic phospholipids employing different techniques. In this sense, Colqui et al. combined the surface pressure–molecular area and cyclic voltammetry techniques to demonstrate that TFPZ is partitioned in a DSPG monolayer adsorbed at air/water and water/1,2-dichloroethane interfaces, changing the structure of the film. The authors observed a fluidizing effect, dependent on the time and the drug concentration, which can be produced either from the organic or from the aqueous phase due to the amphiphilic nature of this drug, leading to an increment in the permeability of the film. Nevertheless, the expansion of the lipid layer is enhanced when TFPZ acts from the aqueous phase [12].

Apart from drug–lipid monolayer interaction studies, special attention has been paid to interaction studies between nanoparticles and this type of film due to the increasing applications of nanoparticles in general [1,19]. However, there are not many articles related to the effect of magnetic nanoparticles (MNPs) on phospholipid films compared to the large number of articles that propose their use in biomedicine [20]. Among the different MNPs, magnetite Fe_3_O_4_ is the most widely proposed for use in biomedicine, as it appears to be the most biocompatible. MNPs are coated with organic materials to increase biocompatibility, prevent aggregation and also to be dispersible in aqueous media [20,21,22]. Carbohydrates are good candidates for coating MNPs, since they have been proposed in several articles [1]. The basic characteristics of the material for medical applications are based on determining the affinity of cells to their surface, which, in turn, depends mainly on the chemical nature and topography of the biomaterial. Therefore, it is important to analyze the physicochemical properties of MNPs with different coated materials generally proposed to be used in biomedicine.

In this manuscript, we aim to explain and characterize the interactions between polymer-coated MNPs and the cell membrane interface at the molecular level. For this purpose, we analyzed the interaction between phospholipids films with MNPs coated with two polysaccharides, chitosan (CHI) and diethylaminoethyl dextran (DEAE-D). Both polymers are made up of glucose units. [20]. CHI is a biopolymer with a linear structure of β-(1,4)-2-amino-deoxy-D-glucopyranose repeating units, characterized by the presence of an amine moiety located on the second carbon atom in the polysaccharide chain and by hydroxyl groups located at C3 and C6 atoms. It is mainly obtained by deacetylation of the amino groups in chitin, but also by isolation from the exoskeleton of marine crustaceans and some fungi [23]. In recent years, CHI-based composite materials have gained wider use in medicine for drug delivery, tissue engineering and wound healing [23]. DEAE-D is also a biopolymer of glucose with -1,6 glycosidic linkages but also it has ramifications in its structure, beginning from the -(1-4)-linkage (as well as -1,2 and -1,3 linkages) [17]. DEAE-D contains three kinds of charged amine groups. The structures of CHI and DEAE-D are shown in Figure 1. It is important to highlight that in this study neither in vivo nor in vitro tests were performed to evaluate the interaction of the MNPs:drug compound with cells. Although cell membranes are largely composed of phospholipids, they also contain proteins and carbohydrates. Therefore, the limitation of the technique used in this work is that the affinity of the MNPs:drug compounds with real cells could differ slightly from that observed in this work, since only phospholipids are taken into account.

One of the most often described indirect method for testing the biocompatibility of nanoparticles and microparticles is using a biomimetic model of biological membranes. Among the proposed models, the Langmuir film is the most popular. The effect of solid nanoparticles on the properties of Langmuir phospholipid monolayers is a topic of increasing interest because such systems may be applied as models in several fields related to biological systems, biomembrane response or respiratory physiology [24].

In this work, the binding properties and insertion of MNPs:Drugs on a Langmuir film prepared from 1,2-distearoyl-sn-glycero-3-phosphate (sodium salt) (DSPA) is studied using the monolayer technique. The effect of ionic drugs HTFPZ^+^, DCFN^−^ and mixtures of drugs/solid MNPs (Fe_3_O_4_@CHI or Fe_3_O_4_@DEAE-D) on the surface behavior of a DSPA monolayer is examined in detail by a combination of surface pressure (π)–area (A) isotherms, compressibility modulus (Cs^−1^) and also by insertion experiments of drugs or MNPs:Drugs mixtures on phospholipid monolayers, determining the maximum insertion pressure (MIP) values.

## 2. Materials and Methods

### 2.1. Materials

The Fe_3_O_4_@CHI and Fe_3_O_4_@DEAED-D MNPs used in this study are Fe_3_O_4_ MNPs with a core diameter of 12 nm and capped with CHI (Sigma Aldrich, low molecular weight) or DEAE-D (CarboMer, Inc. medium molecular weight). These MNPs were synthesized and characterized as previously reported [21]. Phospholipid DSPAs were purchased from Avanti Polar Lipids (Alabaster, Al). LiCl (subphase electrolyte) was purchased from Sigma Aldrich (p.a. grade), Chloroform and Methanol were acquired from Dorwill. Pharmaceutical drugs, DCFN and TFPZ are from sigma Aldrich. All the reactants used were of the highest commercial purity available. The aqueous solutions were prepared with ultrapure deionized water (MQ).

The lipid solutions were prepared in chloroform:methanol 2:1 *v*/*v* at a concentration value equal to 1.0 mM. Capped MNPs have been dispersed (1 mg mL^−1^) in ethanol.

### 2.2. Methods

#### 2.2.1. Surface Pressure—Area Isotherms

The surface pressure–area isotherms have been measured with a KSV Langmuir balance (Mini-trough II, Helsinki, Finland), using the Wilhelmy method with a platinum plate. A rectangular Teflon trough with an area of 364 mm × 75 mm equipped with two moving barriers seated on the upper edge of the trough was used. The lipid monolayers (50 µL DSPA), HTFPZ^+^:lipid or DCFN^−^ and MNPs:lipid monolayers (3, 12 or 15 µL dispersion of the MNPs + 50 µL lipid) were performed by spreading each mixture at the surface of 10 mM LiCl, 0.01 mM HTFPZ^+^ or DCFN^−^ subphase, pH = 6.00, using a Hamilton micro syringe. About 10 min were allowed for the solvent evaporation before starting the compression. The speed of the compression was 10 mm min^−1^ while the automatic measurement of the lateral surface pressure (π) was carried out. The bare LiCl surface was proven to be clean by compression before each measurement, verifying that the surface pressure value was lower than 0.2 mN m^−1^.

In order to analyze the elastic behavior of the films, the interfacial elastic modulus was calculated as:(1)$${\mathrm{Cs}}^{-1}=-A{\left(\frac{\partial \pi }{\partial A}\right)}_{T}$$
where Cs^−1^ is the compressibility and π is the surface pressure measured at each area (A) point of the surface pressure–area isotherm [25,26].

#### 2.2.2. Insertion Experiments

Maximum insertion pressure (MIP) experiments were realized on a teflon mini trough with a constant surface area (A = 15.8 cm^2^). The trough was filled with 5 mL of LiCl 10 mM. The lipids were spread at the surface to reach a surface pressure in the range of 8 and 30 mN m^−1^. Then, successive injections of HTFPZ^+^ or DCFN^−^ and MNPs:HTFPZ^+^ or DCFN^−^ were made underneath the lipid monolayer, as shown in Figure 2b, and the final pressure value after 30 min was determined. The surface pressure–time curves were evaluated by establishing the initial surface pressure (π_i_) and the change in surface pressure after the injections of the MNPs (Δπ = π_f_ − π_i_). Δπ data are plotted with respect to the initial pressure (π_i_) and synergy, and maximum insertion pressure values are obtained, which give information of the MNPs–lipid interaction, as previously reported [17].

## 3. Results

### 3.1. Langmuir Monolayers

#### 3.1.1. HTFPZ^+^:DSPA and DCFN^−^:DSPA Composite Langmuir Films

Molecular interactions between pharmaceutical drugs and membranes can be studied with the Langmuir approach by recording the surface pressure–area (π–A) isotherms during compression of films containing components in different proportions. In this sense, the effect of HTFPZ^+^ and DCFN^−^ on the anionic DSPA monolayers was analyzed by surface pressure isotherms, injecting mixtures of DSPA and HTFPZ^+^ (HTFPZ^+^:DSPA) or DCFN^−^ (DCFN^−^:DSPA), as shown in Figure 1. For both mixtures, drugs:DSPA, the response is different. In these figures, the area is given in terms of area per lipid molecule and therefore any change from pure DSPA can be attributed to the interaction with drug molecules.

For the π–A curve recorded at a minimum ratio drug/lipid, HTFPZ^+^:DSPA 15 mol%, Figure 1a, at low surface pressure, below 10 mN m^−1^, it can be seen that the isotherm is shifted towards larger areas per lipid molecule in comparison with the curve obtained only with DSPA. Thus, the lift-off of the surface pressure started at a larger area per molecule. At surface pressures above 10 mN m^−1^, the isotherm shifts to lower areas. The same behavior can also be observed for 55 and 96 mol% mixture monolayers, but with the difference that for these ratios, the shift towards lower areas is even more important.

For the mixture monolayer HTFPZ^+^:DSPA 96 mol%, in particular, important changes in the surface pressure–area isotherm can be noted. A change in the shape of the isotherm is evident, appearing as a plateau after the lift-off, corresponding to the liquid expanded–liquid condensed phase transition, at approximately 17 mN m^−1^. Similar behavior was previously reported for HTFPZ^+^ using DSPG monolayers [12]. For all curves, pure DSPA and 15, 55 and 96 mol% HTFPZ^+^:DSPA, the collapse pressure is similar.

Figure 1b shows the response of 15 and 90 mol% DCFN^−^:DSPA mixed monolayer isotherms. As can be seen, no important changes in the shape of the pure DSPA isotherm can be observed, except for a slight shift of the curve towards lower areas, which became more important at a higher molar ratio. Additionally, the lift-off of the surface pressure–area isotherm started at a lower area per molecule. It is worth mentioning that pure HTFPZ^+^ and DCFN^−^ do not form a Langmuir monolayer, as the maximum surface pressure upon compression is around 0.6 mN m^−1^, as shown in the light blue curves in Figure 1a,b, indicating that both drugs are not surface active.

#### 3.1.2. Surface Pressure Isotherms of MNPs:DSPA Containing HTFPZ^+^ in the Subphase

The effect of HTFPZ^+^ on a pure DSPA monolayer and on mixt MNPs/DSPA monolayers with different amounts of Fe_3_O_4_@CHI or Fe_3_O_4_@DEAE-D MNPs (Fe_3_O_4_@CHI:DSPA or Fe_3_O_4_@DEAE-D:DSPA) can be noted on surface pressure–area compression isotherms shown in Figure 2a,b, respectively. Additionally, for these figures, the area is given in terms of area per lipid molecule. The complete characterization of these MNPs was previously reported, where the size of both MNP types was shown to be around 12 nm [21,22].

The DSPA isotherm using LiCl 10 mM as the electrolyte in the subphase shown in Figure 2 (black line) displays the well-known profile with a phase transition from the gaseous to the liquid condensed phase, consistent with values previously reported [22].

However, when HTFPZ^+^ is present at the subphase, important changes in the surface pressure–area isotherms are visible. Essentially, the isotherm shows the same behavior observed for an HTFPZ^+^:DSPA monolayer. For lower surface pressures, around 10 mN m^−1^, the isotherm shifts towards larger areas per lipid molecule, indicating that HTFPZ^+^ can be incorporated from the subphase into the film and a consequent film expansion occurs, as shown schematically in Figure 2a left. Additionally, the same change in the shape of the isotherm is evident, appearing a plateau after the lift-off, corresponding to the liquid expanded–liquid condensed phase transition, at approximately 10 mN m^−1^. At higher surface pressures, the isotherm shifts to a lower area per molecule, which can be attributed to the expulsion of some DSPA molecules attached to the HTFPZ^+^ molecules previously incorporated at the monolayer. Another interesting feature in the isotherm of DSPA in the presence of HTFPZ^+^ in the subphase is the lowering of the collapse pressure, from 49.41 to 31.1 mN m^−1^, with a mean molecular area per molecule of 30 A^2^.

For MNPs:DSPA mixtures monolayers when HTFPZ^+^ is present at the subphase, the same shape of the surface pressure–area isotherms is observed. In both kind of films, Fe_3_O_4_@CHI:DSPA or Fe_3_O_4_@DEAE-D:DSPA, Figure 2a,b, respectively, the surface pressure isotherm is also shifted towards larger area per lipid molecule, but with the principal difference that the HTFPZ^+^ can be incorporated up to even higher surface pressures, around 20 mN m^−1^, indicating that in these hybrid films, the HTFPZ^+^ can be incorporated more easily from the subphase.

Additionally, it is important to highlight that in the presence of MNPs in the monolayer, the plateau corresponding to the liquid expanded–liquid condensed phase transition occurs at a larger surface pressure, at approximately 17 mN m^−1^, for both kinds of films. This change in the surface pressure for the phase transition can be attributed to the presence of MNPs at the monolayer that can intercalate between the molecules of lipids, making the expanded liquid phase more stable. Furthermore, the collapse pressure increases from 31.1 mN m^−1^ for DSPA with HTFPZ^+^ at the subphase to 42.5 mN m^−1^ for MNPs:DSPA hybrid films with HTFPZ^+^ at the subphase, indicating that the presence of these MNPs make the monolayer more stable.

Additionally, the presence of hybrid films at the air/water interface increases the collapse pressure from 31.1 mN m^−1^ for DSPA with HTFPZ+ in the subphase to 42.5 mN m^−1^ for MNP:DSPA hybrid films, indicating that the presence of this MNP makes the monolayer more stable compared to the pure DSPA monolayer with HTFPZ+ in the subphase.

In addition, for Fe_3_O_4_@CHI:DSPA, Figure 2a, a slight shift of the isotherms towards smaller areas can be observed. This shift is even more remarkable when increasing the amount of MNPs at the monolayer. This behavior is probably due to the removal of some lipid molecules, together with MNPs, out of the monolayer, since, in this figure, the area is given in terms of area per lipid molecule. For Fe_3_O_4_@DEAE-D:DSPA, Figure 2b, this shift towards smaller areas can also be observed, but to a lesser extent, indicating that interactions between Fe_3_O_4_@CHI and DSPA are stronger than for Fe_3_O_4_@DEAE-D.

Additionally, compression modulus values, Cs^−1^, were calculated as mentioned in Section 4. The Cs^−1^ value provides information about monolayer properties such as fluidity, therefore, provides a quantitative measurement of the monolayer in plane packing elasticity and allows for classifying the state of the monolayer as: liquid-expanded (Cs^−1^ = 10–100 mN m^−1^), liquid-condensed (Cs^−1^ = 100–250 mN m^−1^) and condensed (Cs^−1^ > 250 mN m^−1^) [27,28]. Table 1 shows the values of Cs^−1^ obtained at a constant pressure equal to 30 mN m^−1^. The value of the elastic area compressibility modulus to pure DSPA lipids is around 275 mN m^−1^, which corresponds to a condensate phase [27]. The incorporation of HTFPZ^+^ into DSPA monolayer significantly decreases Cs^−1^, changing the monolayer to a liquid-expanded state, with a Cs ^−1^ value equal to 100 mN m^−1^. For the MNPs:lipid hybrid monolayer, the general trend observed for both kinds of MNPs is a marked decrease in the Cs^−1^, indicating that the films are more compressible and acquired more liquid-expanded character in reference to the pure lipid. A similar behavior was previously observed for nanoparticles of Ag by Villanueva et al. [29]. Appendix A shows the changes in Cs^−1^ of DSPA monolayers at different MNP volumes for mixed monolayers, where significant differences are evident. For MNPs:DSPA monolayers, the Cs^−1^ displayed a minimum value at π = 17 mN m^−1^, which was the phase transition point induced for the insertion of HTFPZ^+^ into the monolayer and confirmed the results of surface pressure–area isotherms.

#### 3.1.3. Surface Pressure Isotherms of MNPs:DSPA Containing DCFN^−^ in the Subphase

Figure 3 shows the surface pressure–area compression isotherms of pure DSPA and with increasing amounts of MNPs forming a mixed film of Fe_3_O_4_@CHI:DSPA or Fe_3_O_4_@DEAE-D:DSPA, Figure 3a,b respectively, in the absence or presence of DCFN^−^ at the subphase. Here, the area of the figure is also given in terms of area per lipid molecule and therefore any change from pure DSPA can be attributed to the interaction with MNPs or drug molecules.

The black line corresponds to a DSPA isotherm using LiCl 10 mM as the electrolyte in the subphase. The curves also display the well-known profile of pure DSPA. The collapse occurs at 48.5 mN m^−1^, with a mean molecular area of 37 A^2^ per molecule. When DCFN^−^ is present at the subphase, no significant changes in the surface pressure–area isotherms of DSPA are visible, except for a slight shift of the curve towards larger areas, indicating that DCFN^−^ has been incorporated into the film. A similar behavior was previously reported by S.M.B. Souza et al. for diclofenac in soya phosphatidylcholine, and the authors also reported that with the increasing diclofenac concentration at the subphase, the isotherm becomes more expanded and more compressible [6].

No important changes in the surface pressure–area isotherms are observed when MNPs:DSPA mixtures are spread on the subphase. For Fe_3_O_4_@CHI:DSPA, Figure 3a, a slight shift of the curves towards larger areas can be seen at lower pressures, under 20 mN m^−1^, for 12 µL and 15 µL Fe_3_O_4_@CHI:DSPA hybrid films. However, at higher pressures, above 20 mN m^−1^, the curves shift towards smaller areas. This behavior indicated that at lower pressures, DCFN- is incorporated into the hybrid films from the subphase (Figure 2a left), but when the surface pressure increases, DCFN- is expelled from the monolayer towards the subphase, together with some MNP:lipid molecules (Figure 2a right). Figure 3b shows the isotherms for Fe_3_O_4_@DEAE-D:DSPA hybrid films where a similar behavior can be observed. At low pressures, under 25 mN m^−1^, a slight expansion of the isotherms for 12 µL and 15 µL Fe_3_O_4_@CHI:DSPA occur, while the changes in the isotherms are practically negligible at high pressures, denoting that the drug previously incorporated into the monolayer is then expelled from the interface to the subphase.

Regarding the compression modulus values, the incorporation of DCFN^−^ into a pure lipid monolayer does not affect the state of the monolayer, with a Cs^−1^ value of 261.09 mN m^−1^, as shown in Table 1. For the Fe_3_O_4_@CHI:DSPA hybrid monolayer, the Cs^−1^ value decreases with the increasing amount of MNPs, from 161.51 mN m^−1^ for a 3 µL Fe_3_O_4_@CHI:DSPA mixture to 115.53 mN m^−1^ for a 15 µL Fe_3_O_4_@CHI:DSPA hybrid film. These values also demonstrate that the hybrid monolayers are more compressible and acquired a more liquid-expanded character in reference to the pure lipid. For the 3 µL Fe_3_O_4_@DEAE-D:DSPA hybrid film, the Cs^−1^ value is the highest, 442.99 mN m^−1^, demonstrating that the condensed state of the monolayer is preserved for this mixture film. Besides increasing the amounts of Fe_3_O_4_@DEAE-D MNPs at the film, the monolayer acquired a liquid-condensed state. Appendix A shows the changes in Cs^−1^ of DSPA monolayers at different MNPs volumes for a mixed monolayer with DCFN^−^ at the subphase. For these experiments, no important change in the shape of Cs^−1^ can be observed.

### 3.2. Insertion Experiments on DSPA Pre-Formed Monolayer

#### 3.2.1. Insertion of HTFPZ^+^ or DCFN^−^ with Preformed DSPA Monolayer

These measurements were carried out with the aim of studying the degree of permeation of these drugs across the DSPA monolayer. These insertion experiments were performed as described in Section 4. The surface pressure in living cells is in the range of 20–30 mN m^−1^, depending on the type and location of the cell [20]. Due to this, the adsorption experiments were carried out using an initial surface pressure of DSPA of π_i_ = 30 mN m^−1^ and different HTFPZ^+^ or DCFN^−^ concentrations in the subphase, and the results are shown in Figure 4a. In the case of DCFN^−^, no effect on the surface pressure values is observed when the drug is injected in the subphase, even for high concentrations of the DCFN^−^ at the subphase, such as 5.5 mM, where the change in the surface pressure, ∆π = π_f_ − π_i_, is only 1.6 mN m^−1^, indicating that, for this drug, the insertion into a preformed monolayer of DSPA does not occur. On the contrary, for HTFPZ^+^, at a concentration of 0.3 mM, a ∆π = 7.0 mN m^−1^ is detected and reaches a value of ∆π = 8.5 mN m^−1^ for a concentration of 1 mM. This indicates that the HTFPZ^+^ can insert into the preformed monolayer of DSPA. Similar observations were previously reported for HTFPZ^+^ by Colqui et al. using DSPG and by A.A. Hidalgo for dipalmitoyl-phosphatidyl-choline (DPPC) or dipalmitoyl-phosphatidyl-glycerol (DPPG) monolayers [12].

Since the insertion behavior of the HTFPZ^+^ and DCFN^−^ is characterized in function of the concentration, a selected concentration of 0.5 mM for HTFPZ^+^ and 1.5 mM for DCFN^−^ were used for MIP experiments. The change in the surface pressure of the DSPA model membrane was followed during the interaction with HTFPZ^+^ and DCFN^−^ in the subphase. The interaction of the drugs with DSPA monolayers are studied at different compactness. The plot of the change in the surface pressure (Δπ) versus the initial surface pressure (π_i_) gives a linear relationship, x tangent intercept is the MIP and is shown in Figure 4b. For HTFPZ^+^, at all the different initial surface pressures used, an increase in the final surface pressure is observed due to interactions between the drug and the phospholipid monolayer. Lower π_i_ produces a big increase in the pressure and higher π_i_ surface pressure results in a smaller increase in the surface pressure. The MIP is the value of surface pressure above which no incorporation can occur. As the surface pressure in living cells is in the range of 20–30 mN m^−1^, MIP values higher than 30 mN m^−1^ suggest that drug penetration is expected in living systems. For HTFPZ^+^, the MIP is 35.6 mN m^−1^, so the insertion of this drug into membrane cells can be expected. Nevertheless, for DCFN^−^, any changes at all π_i_ can be observed, which leads us to conclude that this pharmaceutical drug cannot insert into a DSPA preformed monolayer.

#### 3.2.2. Insertion of MNPs:HTFPZ^+^ or MNPs:DCFN^−^ with Preformed DSPA Monolayer

Magnetic nanoparticles such as Fe_3_O_4_ are widely used in nanobiomedicine for drug delivery applications [30]. While Y. Ding et al. use MNPs of polymerized-chitosan coated Fe_3_O_4_ for drug delivery of the anticancer drug 5-fluorouracil [31], S. Uribe and collaborators develop an MNP of Fe_3_O_4_ coated with silica for an ibuprofen loading and release model [32] and Y. Jia et al. also report the use of MNPs of Fe_3_O_4_ for doxorubicin drug delivery [33]. Due to all this background research, we decide to probe the insertion of mixtures of MNPs:HTFPZ^+^ or DCFN^−^ into the preformed DSPA monolayer and the results are shown in Figure 5. Additionally, these experiments were performed as described in Section 4, and the schematic representation of the experiments is shown in Figure 2b. Briefly, 250 µL of MNPs dispersion was mixed with solutions of HTFPZ^+^ or DCFN^−^ to reach a final concentration of 0.5 mM and 3.5 mM, respectively, in the subphase. Mixing MNPs and drug before injecting under the DSPA monolayer can improve the interactions. MNPs:HTFPZ^+^ mixtures insertion experiments are presented in Figure 5a. For Fe_3_O_4_@CHI:HTFPZ^+^ and Fe_3_O_4_@DEAE-D:HTFPZ^+^ mixtures, an increase in the MIP value from 35.6 mN m^−1^ for HTFPZ^+^, to 40.2 and 42.5 mN m^−1^ for Fe_3_O_4_@DEAE-D:HTFPZ^+^ and Fe_3_O_4_@CHI:HTFPZ^+^, respectively, can be observed. This increase in the MIP value leads us to conclude that the use of MNPs can improve the insertion of the drug into DSPA preformed monolayers, reaching values even higher than 30 mN m^−1^, the expected value of the surface pressure in living cells. In our previous work, we reported that HTFPZ^+^ can interact strongly with polysaccharides-coated MNPs through hydrophobic interactions [22]. Due to this, it is expected that in the mixtures, the MNPs are interacting with HTFPZ^+^, improving the insertion of the drug drive by hydrophobic interactions between MNPs and HTFPZ^+^. Additionally, it is important to mention that these MNPs are positively surface charged, due to the coating with cationic polymers. Previous works reported that nanoparticles with a positive surface charge were found to cross the bilayer membrane faster than particles with a negative surface [20,34], so the surface charge of the MNPs is also an important property which can influence the insertion process.

Figure 5b shows the MNPs:DCFN^−^ mixtures insertion experiments. Here it is important to highlight the fact that for DCFN^−^, no insertion activity was observed at any concentration of drug and any π_i_ of the DSPA preformed monolayer. As can be observed, for both kind of mixtures, Fe_3_O_4_@CHI:DCFN^−^ and Fe_3_O_4_@DEAE-D:DCFN^−^, the MIP reach an impressive value of 39.3 mN m^−1^ for Fe_3_O_4_@CHI:DCFN^−^ and the MIP value is even higher for the Fe_3_O_4_@DEAE-D:DCFN^−^ mixture, reaching an 67.1 mN m^−1^. This important increase in the MIP value leads us to conclude that DCFN^−^ can be adsorbed only in mixed films. Additionally, we previously reported that DCFN^−^ can interact strongly with MNPs driven by electrostatic interactions. In this same work, we also use DFT calculations to calculate the binding energy of a DCFN^−^ anion with ammonium cations present in the polymers structures that are covering the MNPs, concluding that the amines group in DEAE-D interacts stronger than CHI [22]. All of this indicates that DCFN^−^ is strongly interacting with Fe_3_O_4_@DEAE-D and can improve the drug adsorption into DSPA preformed monolayers. On the other hand, in the case of HTFPZ^+^, as it is a cationic drug, hydrophobic interactions are important, but these are no stronger than electrostatic interactions and due to this, the insertions are not as favorable as in the case of DCFN^−^.

From the results shown, we can conclude that the use of mixture MNPs:drug can improve the drug penetration or insertion from an aqueous phase into the DSPA monolayer. This insertion is more important for MNPs:DCFN^−^, reaching high MIP values. Two important parameters can be calculated from the Δπ versus π_i_ plot: MIP and synergy. The synergy (slope of + 1 of the linear fitting of Δπ versus π_i_) relates the affinity of the MNPs to the monolayer. If synergy > 0, the affinity between MNPs and monolayer is positive, but if synergy < 0, the MNPs have less affinity to the monolayer [20]. With the aim of summarizing the results obtained from insertion experiments, Figure 6a shows the MIP for all MNPs:drug mixtures and Figure 6b shows the synergy.

For MIP values, the results are diverse, but always, MIP values are higher when the drug is interacting with MNPs. The purple band in the range of 20–30 mN m^−1^ shown in Figure 6a represents the compactness in biological membranes. For all the studied systems, MNPs:drugs reach this region and even achieve much higher MIP values. In the case of synergy, there is not much difference for the different MNPs:drug, except for Fe_3_O_4_@DEAE-D:DCFN^−^. For this system, the synergy value is higher compared to the rest of the mixtures, which is in agreement with the value of MIP, that is the higher value, which means that this MNP:drug mixture can be incorporated into the most compact membrane. All of this indicates a favorable condition for incorporation.

## 4. Discussion

As can be observed in all figures that pure DSPA blank isotherms, shown in black line, display the well-known profile with phase transition from the gaseous to the condensed liquid phase. The monolayer collapse is evident at a surface pressure of 48.7 mN m^−1^, with a mean molecular area of 35 A^2^ per molecule, consistent with values previously reported [22].

For all the π–A curves performed with drug/lipid mixtures, the shift to larger areas per lipid molecule at lower 10 mN m^−1^ indicates that HTFPZ^+^ is incorporated into the film, producing film expansion. However, at surface pressures above 10 mN m^−1^, the isotherm shifts to lower areas, indicating that the HTFPZ^+^ molecules, previously incorporated into the DSPA monolayer, are expelled out of the monolayer along with some lipid molecules from the monolayer. For the increasing amount of HTFPZ^+^ into the film, the shift to lower areas at higher pressures becomes more evident. This fact is due to more molecules of lipid being expelled out of the monolayer at the air/water interface into the subphase, demonstrating that at a higher HTFPZ^+^ concentration, electrostatic interactions between the cationic drug and the anionic phospholipid are more favorable. In addition, a strong hydrophobic interaction between HTFPZ^+^ and organic molecules was previously reported [22]. For all isotherms, pure DSPA and 15, 55 and 96 mol% HTFPZ^+^:DSPA, the collapse pressure is similar, indicating that the presence of the HTFPZ^+^ molecules does not disturb the stability of the monolayer.

Additionally, for the DCFN^−^:DSPA mixed monolayer isotherm, slight shifts of the curves towards lower areas were observed, being more evident at higher molar ratios due to the expulsion of lipid molecules from the monolayer to the subphase. It is worth mentioning that the shifts produced in this case are smaller than in HTFPZ^+^ mixture films, since DCFN^−^ and DSPA interactions are weaker due to the fact that these molecules can interact only through hydrophobic interactions. Additionally, recently, it was demonstrated that the hydrophobic interactions between DCFN^−^ and organic polymers are weaker than in the case of HTFPZ^+^ and the same polymers [22].

The same behavior for the different types of experiments, HTFPZ^+^: lipid hybrid films and lipid films with HTFPZ^+^ at the subphase, was observed by isotherm experiments. The principal difference is that for the second type of experiment, the collapse pressure decreases from 49.41 mN m^−1^ for pure DSPA to 31.1 mN m^−1^. This important decrease can be attributed to the fact that the incorporation of the cationic drug into the film decreases the stability of the monolayer, anticipating the 2D–3D process, as previously reported for other pharmaceutical drugs, such as ibuprofen [28], paclitaxel [35] and fluoxetine [36]. These results also reinforce what was previously mentioned above, that part of DSPA molecules was expelled from the monolayer at higher surface pressure, as represented in Figure 2a right.

HTFPZ^+^ can be incorporated into MNPs:DSPA mixtures monolayers at even higher surface pressures, around 20 mN m^−1^, indicating that in these hybrid films, the drug can be incorporated more easily from the subphase. For this hybrid film, the phase transition is also observed, but at higher surface pressure, which can be attributed to the fact that the presence of MNPs at the monolayer can intercalate between lipid molecules, making that transition more difficult. Furthermore, the presence of hybrid films at the air/water interface increases the collapse pressure from 31.1 mN m^−1^ for DSPA with HTFPZ^+^ at the subphase to 42.5 mN m^−1^ for MNPs:DSPA hybrid films, indicating that the presence of these MNPs make the monolayer more stable. The principal difference between Fe_3_O_4_@CHI:DSPA and Fe_3_O_4_@DEAE-D:DSPA hybrid films is that, for the MNPs covered with CHI, the shifts towards smaller areas are even more remarkable for increasing amounts of Fe_3_O_4_@CHI MNPs at the monolayer. This indicates that for these kind of MNPs, the interaction between MNPs and lipids are stronger.

When DCFN^−^ is present at the subphase, no significant changes in the surface pressure–area isotherms of MNPs:DSPA films are visible, except for a slight shift under 20 mN m^−1^ towards larger areas for 12 µL and 15 µL MNPs:DSPA hybrids films, indicating that DCFN^−^ has been incorporated into the film. Additionally, at higher pressures, above 20 mN m^−1^, the curves shifted towards smaller areas. This behavior indicated that, at lower pressures, the DCFN^−^ is incorporated into the hybrid films from the subphase, Figure 2a left, but when the surface pressure increases, the molecules, along with some MNPs:lipids molecules, are expelled from the monolayer to the subphase, Figure 2a right.

Insertion experiments inform about the positive or negative interactions between pharmaceutical drugs or MNPs/drug composite with preformed model lipid membranes. It has been shown that DCFN^−^ cannot adsorb into a preformed monolayer of DSPA, probably due to electrostatic repulsion between negatively charged DCFN^−^ and anionic lipid. Otherwise, HTFPZ^+^ can insert into a preformed DSPA monolayer. The MIP is the value of surface pressure above which no incorporation can occur. As the surface pressure in living cells is in the range of 20–30 mN m^−1^, MIP values higher than 30 mN m^−1^ suggest that drug penetration is expected in living systems. For HTFPZ^+^ the MIP is 35.6 mN m^−1^, so the insertion of this drug into living cells can be expected.

The results obtained by insertion experiments carried out with MNPs:HTFPZ^+^ or MNPs:DCFN^−^ demonstrate enhanced adsorption of HTFPZ^+^ into a DSPA preformed monolayer, increasing the MIP value from 35.6 mN m^−1^ for HTFPZ^+^ to 40.2 and 42.5 mN m^−1^ for Fe_3_O_4_@DEAE-D:HTFPZ^+^ and Fe_3_O_4_@CHI:HTFPZ^+^, respectively. Moreover, the DCFN^−^ reaches an impressive MIP value of 39.3 mN m^−1^ for Fe_3_O_4_@CHI:DCFN^−^ and an even higher MIP value for the Fe_3_O_4_@DEAE-D:DCFN^−^ mixture, reaching 67.1 mN m^−1^. This important increase in the MIP value leads us to conclude that MNPs can act as carriers for drug delivery of these pharmaceutical drugs, especially for DCFN^−^.

This work is focused on describing the interactions between MNPs and different pharmaceutical drugs and how these composites can be inserted into preformed monolayers. The next step in the research is to study the influence of an external magnetic field of varying intensity on the insertion of these composites into preformed phospholipid monolayers.

## Data Availability

Not applicable.

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
