# Peer review of "Interaction between Pharmaceutical Drugs and Polymer-Coated Fe3O4 Magnetic Nanoparticles with Langmuir Monolayers as Cellular Membrane Models"

_pharmaceutics, 2023, doi:10.3390/pharmaceutics15020311_

Round 1

Reviewer 1 Report

The authors have done remarkable job in studying the interactions between MNPs and different pharmaceutical drugs and how these composites can be inserted into preformed monolayers. 

Author Response

 The authors thank the reviewer for the review and for the positive comment about the manuscript. 

Reviewer 2 Report

The manuscript is well-designed and well-written. My comments are:

-          Kindly provide relevant references for the equations used and ensure all abbreviations are defined at first appearance.

-          It is highly encouraged to re-number different subheadings for section 2.

-          The quality of figure 2 is not acceptable. Please replace it with a more transparent graphic pictorial.

-          In the abstract, the authors must elaborate on the application of the most recent nanoparticle-based systems for drug delivery by discussing the articles below. DOI: 10.1016/j.molliq.2021.118271 - 10.1016/j.jddst.2022.103459 - 10.1021/acsomega.1c03816

-          There is no data concerning the size of NPs.

-          The authors performed multiple assays to ensure interaction between the drug, polymer-coated NPs, and Langmuir monolayers. However, they did not perform biological experiments to evaluate the effect of such formulation using in vivo and in vitro tests. Thus, I suggest revising the manuscript's title and removing the drug delivery part, as there is no proof that the developed nanosystem would act as an efficient drug delivery platform. Also, this issue must be mentioned as a limitation of the study.

-          Other limitations of the study must be discussed in detail.
I would like to review the revised version of the manuscript.

Author Response

The manuscript is well-designed and well-written. My comments are:

-          Kindly provide relevant references for the equations used and ensure all abbreviations are defined at first appearance.

R./ The authors thank the reviewer for the revision of the paper, and for their suggestions to improve the article.  In this new version of the manuscript, we included relevant references for the equations.

-          It is highly encouraged to re-number different subheadings for section 2.

R./ The authors agree with the reviewer's considerations, so we renumber section 2.

-          The quality of figure 2 is not acceptable. Please replace it with a more transparent graphic pictorial.

R./ We agree with the reviewer's consideration, in this new version of the manuscript the quality of figure 2 was enhanced.

-          In the abstract, the authors must elaborate on the application of the most recent nanoparticle-based systems for drug delivery by discussing the articles below. DOI: 10.1016/j.molliq.2021.118271 - 10.1016/j.jddst.2022.103459 - 10.1021/acsomega.1c03816

R./ We agree that these articles are interesting so we included them in the manuscript.

-          There is no data concerning the size of NPs.

R./ In section 4, in the ´´Materials´´ part were specified the core diameter of the MNPs. The complete characterization of these NPs was previously reported (cite 21 and 22 in the manuscript). Here we introduce a paragraph mentioning the size of the nanoparticles.

-          The authors performed multiple assays to ensure interaction between the drug, polymer-coated NPs, and Langmuir monolayers. However, they did not perform biological experiments to evaluate the effect of such formulation using in vivo and in vitro tests. Thus, I suggest revising the manuscript's title and removing the drug delivery part, as there is no proof that the developed nanosystem would act as an efficient drug delivery platform. Also, this issue must be mentioned as a limitation of the study.

R./ We apologize to the reviewer and in this new version of the manuscript we changed the title ´´ Interaction between pharmaceutical drugs and polymer coated-Fe3O4 magnetic nanoparticles with Langmuir monolayers as cellular membrane models for drug delivery´´ to´´ Interaction between pharmaceutical drugs and polymer coated-Fe3O4 magnetic nanoparticles with Langmuir monolayers as cellular membrane models´´. Also, this issue was mentioned in this new version of the manuscript.

-          Other limitations of the study must be discussed in detail.
I would like to review the revised version of the manuscript.

R./ Here we introduce a paragraph mentioning the limitation of this study.

Reviewer 3 Report

The article by Betancourt et.al. describing the effect of surface modification of MNPs in the membranes insertion behavior is a well designed study. This reviewer found that the experiments conducted and the data presented are OK for the scope of the manuscript from the authors’ perspectives.

However, this reviewer found that similar studies using exact same surface agents have been conducted on other particles such a small Au and Ag. I would have expected that some studies to establish the magnetic behavior of the system after the insertion experiments would have added more value and significance to this study. 

Major English edits are required. Some of them are noted here such as in lines 48, 76, 255, 290, 446 etc. the English formation and even words used are quite wrong. 

Author Response

The authors thank the reviewer for the review and appreciate his/her positive feedback on the manuscript. 

However, this reviewer found that similar studies using exact same surface agents have been conducted on other particles such a small Au and Ag. I would have expected that some studies to establish the magnetic behavior of the system after the insertion experiments would have added more value and significance to this study.

R/ We agree with the reviewer's consideration. A manuscript is currently being drafted about how the presence of a magnetic field of different intensities affects the incorporation of these MNPs on biological membrane models, including too a kinetic study of these incorporations.

Major English edits are required. Some of them are noted here such as in lines 48, 76, 255, 290, 446 etc. the English formation and even words used are quite wrong.

R/ We apologize to the reviewer and in this new version of the manuscript, major English edits were done.

Reviewer 4 Report

The presented research is well written and presented. Correct in all parts. Authors should check the English language once again. Here is what I have found:

1)     Line 48 (page 2): its -> it

2)     Line 52  (page 2): Introduce what is DCFN

3)     Line 69  (page 2): film -> films

4)     Page 8. The second sentence of the second paragraph should be rewritten.

Well done, once again.

Author Response

The authors thank the reviewer for the review and for the positive comment about the manuscript. We have had the text corrected by a professional.

Round 2

Reviewer 2 Report

The manuscript is well revised and can be accepted for publication.